# A comprehensive two-hybrid analysis to explore the *Legionella pneumophila* effector–effector interactome

Harley O'Connor Mount,[1] Malene L. Urbanus,[2] Dayag Sheykhkarimli,[3,4] Atina G. Coté,[3,4] Florent Laval,[5,6,7,8,9,10] Georges Coppin,[5,6,7,8] Nishka Kishore,[3,4] Roujia Li,[3,4] Kerstin Spirohn-Fitzgerald,[5,6,7] Morgan O. Petersen,[2,11] Jennifer J. Knapp,[3,4] Dae-Kyum Kim,[3,4] Jean-Claude Twizere,[8,9] Michael A. Calderwood,[5,6,7] Marc Vidal,[5,6] Frederick P. Roth,[1,3,4,12] Alexander W. Ensminger[1,2]

ABSTRACT   *Legionella pneumophila* uses over 300 translocated effector proteins to rewire host cells during infection and create a replicative niche for intracellular growth. To date, several studies have identified *L. pneumophila* effectors that indirectly and directly regulate the activity of other effectors, providing an additional layer of regulatory complexity. Among these are "metaeffectors," a special class of effectors that regulate the activity of other effectors once inside the host. A defining feature of metaeffectors is direct, physical interaction with a target effector. Metaeffector identification, to date, has depended on phenotypes in heterologous systems and experimental serendipity. Using a multiplexed, recombinant barcode-based yeast two-hybrid technology we screened for protein–protein interactions among all *L. pneumophila* effectors and 28 components of the Dot/Icm type IV secretion system (>167,000 protein combinations). Of the 52 protein interactions identified by this approach, 44 are novel protein interactions, including 10 novel effector–effector interactions (doubling the number of known effector–effector interactions).

IMPORTANCE   Secreted bacterial effector proteins are typically viewed as modulators of host activity, entering the host cytosol to physically interact with and modify the activity of one or more host proteins in support of infection. A growing body of evidence suggests that a subset of effectors primarily function to modify the activities of other effectors inside the host. These "effectors of effectors" or metaeffectors are often identified through experimental serendipity during the study of canonical effector function against the host. We previously performed the first global effector-wide genetic interaction screen for metaeffectors within the arsenal of *Legionella pneumophila*, an intracellular bacterial pathogen with over 300 effectors. Here, using a high-throughput, scalable methodology, we present the first global interaction network of physical interactions between *L. pneumophila* effectors. This data set serves as a complementary resource to identify and understand both the scope and nature of non-canonical effector activity within this important human pathogen.

KEYWORDS   bacterial effector, host–pathogen, *Legionella pneumophila*, metaeffector, yeast two-hybrid

Many bacterial pathogens actively translocate protein cargo, called "effectors," into host cells to establish a replicative niche. The perspective that bacterial effectors exclusively target host pathways and that effector activity is solely regulated by effector expression or translocation timing has been changing with the discovery of functional effector–effector interactions in several bacterial pathogens (1, 2). The majority of effector–effector interactions described to date are antagonistic but indirect; e.g., one effector antagonizes the action of another effector by acting on host targets. Some

Editor Julia Willett, University of Minnesota Twin Cities, Minneapolis, Minnesota, USA

Address correspondence to Alexander W. Ensminger, alex.ensminger@utoronto.ca.

Harley O'Connor Mount and Malene L. Urbanus contributed equally to this article. Author order was determined alphabetically.

F.P.R. and M.V. are advisors and shareholders of SeqWell, Inc. (Beverly, MA, USA).

See the funding table on p. 18.

indirect antagonists compensate for negative side effects of other effectors (3), while others act on the same host factor targeted by another effector with opposing actions, e.g., adding and removing post-translational modifications (4–10). Effectors in the latter class have the potential to regulate an action on the host in a spatial and/or temporal manner.

Cooperative effector interplay has also been identified. For example, paraeffectors LphD and RomA from *Legionella pneumophila*, an intracellular pathogen that is the causative agent of Legionnaires' disease (11–13), work consecutively by first deacetylating and then methylating K14 of host histone 3 (14). *L. pneumophila* effectors LegC2, LegC3, and LegC7 directly cooperate, forming a soluble N-ethylmaleimide sensitive factor attachment protein receptor (SNARE)-like complex with the human hVAMP4 protein to modulate membrane fusion (15). Another emerging class of effectors, metaeffectors ("effectors of effectors"), do not target host proteins but instead, once inside the host, directly bind and regulate the activity of other effectors (16–20).

*L. pneumophila* (11, 21) uses the Dot/Icm type IVB secretion system (T4SS) to deliver effectors that serve to create a replicative niche in macrophages (22) and protozoan species (23–29). With over 300 effectors (30–32), *L. pneumophila* has diverse types of effector interplay: (i) indirect antagonist effectors (5–9, 33–37), (ii) cooperative paraeffectors (14), and (iii) direct metaeffectors (16, 18, 33, 37–45).

We previously used a genetic interaction screen in the yeast *Saccharomyces cerevisiae* to systematically identify *L. pneumophila* functional effector–effector regulation (18). Altogether, this approach identified 23 effector–effector suppression pairs, in which an antagonist effector suppresses the yeast-growth defect caused by a growth-inhibitory effector. We then examined this subset of effectors for protein–protein interactions (PPIs) using pairwise yeast two-hybrid and mammalian LUMIER to identify nine instances of direct metaeffectors. Collectively, this screen and others have identified 11 confirmed and putative metaeffectors in the *L. pneumophila* effector arsenal (16, 18, 33, 38, 39). However, this is clearly not a complete picture of the possible effector–effector interactions and regulatory pairs in this pathogen: for instance, effectors that do not have a conserved host target in yeast will not inhibit yeast growth and would therefore have been missed in our genetic interaction screen. Similarly, the formation of a SNARE-like complex between LegC2, LegC3, and LegC7 with hVAMP4 (15) is a clear demonstration that not all physical interactions between effectors represent metaeffector–effector regulatory relationships.

To complement our previous genetic interaction screen, we performed a systematic screen for physical effector–effector interactions using the Barcode Fusion Genetic-Yeast Two-Hybrid (BFG-Y2H) assay, a high-throughput Y2H approach with a barcode sequencing readout (46). Because many *L. pneumophila* effectors inhibit yeast growth when overexpressed (18, 47–52), we modified the Y2H vectors to keep them transcriptionally silent prior to the readout of each potential binding event. This inducible BFG-Y2H (iBFG-Y2H) technology was used to screen for protein interactions between 390 effectors and putative effectors and 28 Dot/Icm T4SS components in >167,000 pairwise combinations.

## RESULTS

### Modification of a high-throughput yeast two-hybrid approach to reduce loss of yeast-toxic genes from screening libraries

To screen all *L. pneumophila* effectors for binary physical interactions, we performed a pooled, multiplexed yeast two-hybrid screen that exploits recombinant DNA barcode pairs to detect binary physical interactions within a library of clones (46). Briefly, the BFG-Y2H assay is based on the original Y2H assay (53) in which the yeast transcriptional activator Gal4 is split into a DNA-binding domain (DB) and an activating domain (AD). Each "bait" protein X is fused to DB (DB-X) and each potential "prey" protein Y is fused to the AD domain (AD-Y). A physically interacting pair of proteins X and Y can thus reconstitute the Gal4 protein and drive transcription of reporter genes such as

*GAL1::HIS3*, which allows for growth on medium lacking histidine. A key strength of the BFG-Y2H assay is scalability; Y2H screens are performed in multiplexed pools using Illumina sequencing of molecular barcodes as a readout of the pool composition (46) (Fig. 1A). Each DB-X and AD-Y vector contains a molecular barcode locus with two unique tags ("uptag" and "downtag"), such that one tag is flanked by unique lox sites (*loxP* and *lox2272*). Pools of DB-X and AD-Y barcoded vectors are transformed into yeast cells of an opposite mating type. After these cell pools are mated *en masse*, the subsequent diploid pool is subjected to Y2H selection and control growth conditions. Upon induction of Cre recombinase, lox-flanked tags on reciprocal DB and AD vectors within the same cell are recombined *in vivo*. The resulting chimeric barcodes uniquely identify the DB-X–AD-Y interaction pair, and the abundance of these chimeric barcodes in the sequencing data reflects the abundance of the corresponding strain expressing this interacting pair in the yeast pools (46) (Fig. 1A).

For the purpose of examining physical interactions between *L. pneumophila* effectors, one complication is that standard BFG-Y2H vectors express DB-X and AD-Y using a constitutive *ADH1* promoter, which would express Gal4 domain-fused effector proteins throughout the entire screening process. This is problematic, given that two-thirds of *L. pneumophila* effectors have yeast growth-inhibitory effects that range from severe to mild (18, 47–50, 52, 54–56), which likely reflect the conservation of their targets in *S. cerevisiae* (9, 47, 48, 50, 52, 54–60).

To mitigate the impact of effector-induced growth inhibition, we modified each BFG-Y2H vector by replacing the constitutive *ADH1* promoter with the copper-inducible *CUP1* promoter (61). This allowed us to avoid expressing each DB-X and AD-Y fusion during haploid library construction, expansion, and mating of DB- and AD-containing strains. To confirm inducible expression, we grew yeast with "empty" DB or AD vectors in medium lacking copper to keep expression at a minimum or induced with 1-mM copper for 3, 6, and 24 h. We observed no expression of the AD or DB domain at T0, or when uninduced, and increasing expression at 3, 6, and 24 h (Fig. 1B).

Next, we tested five Y2H PPIs between known growth-inhibitory effectors and their corresponding metaeffectors. These pairs had been identified previously by transforming AD-fused toxic effectors to strains already carrying DB-fused metaeffectors (18). Briefly, we mated AD-fused toxic effectors and DB-fused antagonist haploid strains and selected for diploids (containing both AD and DB vectors on medium lacking copper). We then queried each diploid strain for the ability to grow under two standard Y2H-selective conditions: (i) on medium lacking histidine and (ii) on medium lacking histidine supplemented with 3-AT (an inhibitor of His3p commonly used to increase the stringency of Y2H selection). Of the five pairs, four were captured using the inducible BFG-Y2H vectors (Fig. 1C). In agreement with our previous experiments (18), the strongest interaction pairs LegL1–RavJ and MavE–LegC7/YlfA grew well on both selective conditions, and the weaker interaction pair LupA–LegC3 only supported growth on medium lacking histidine. In a minor deviation from our previous findings using standard Y2H vectors, the fifth metaeffector–effector pair we tested (SidP–MavQ) was only captured under lower stringency conditions (−histidine).

Taken together, these data show that the inducible BFG-Y2H vectors are sufficiently transcriptionally silent to perform the BFG-Y2H screening process of transformation, mating, and diploid selection and the iBFG-Y2H assay captures previously known metaeffector–effector pairs with a similar, but not identical, sensitivity profile.

## An iBFG-Y2H screen of *L. pneumophila* Dot/Icm secretion system components and effectors

To extend this approach to the entire *L. pneumophila* effector arsenal, we combined previously constructed *L. pneumophila* open reading frame (ORF) clones in Gateway vectors (18, 62) with clones for an additional 52 effector and putative effector ORFs, collectively assembling clones for 267 confirmed effectors and 123 putative effectors (Table S1). We also included ORF clones representing 28 components of the Dot/Icm

## A 1. Unique AD and DB tags

AD — uptag | downtag

DB — uptag | downtag

## 2. Mating pool of tagged strains

**AD pool**

**DB pool**

## 3. Grow diploid pool with induction

control          Y2H selection

## 4. Recombine up and down tags

AD up | AD dn

DB up | DB dn

## 5. PCR amplify fused tags

AD up | DB up

AD dn | DB dn

## 6. Illumina sequencing

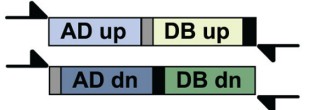

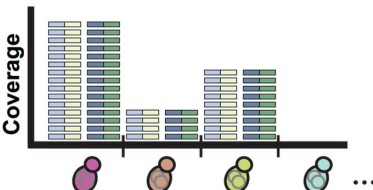

**B**

T0 | 3 h | 6 h | 24 h

kDa  -  | -  + | -  + | -  +   1 mM CuSO$_4$

anti-AD

anti-actin

anti-DB

anti-actin

**C**

AD-EV | AD-effector | AD-EV | AD-effector | AD-EV | AD-effector

$^{DB}$EV-$^{AD}$EV

$^{DB}$LegL1 - $^{AD}$**RavJ**

$^{DB}$LupA - $^{AD}$**LegC3**

$^{DB}$MavE - $^{AD}$**LegC7**

$^{DB}$SdbC - $^{AD}$**SdbB**

$^{DB}$SidP - $^{AD}$**MavQ**

-Leu/Trp | -Leu/Trp/His +CuSO$_4$ | -Leu/Trp/His +CuSO$_4$/3-AT

**FIG 1** Inducible BFG-Y2H. (A) Schematic representation of the iBFG-Y2H screen: (1) All AD-Y and DB-X vectors carry a unique molecular barcode consisting of an uptag and a downtag which one is flanked by lox sites (*loxP* and *lox2272*, gray and black bars); (2) haploid yeast pools with unique barcoded AD-Y and DB-X clones are mated *en masse* and grown on medium lacking tryptophan and leucine to select for diploid cells carrying all combinations of the AD and DB vectors; (3) the diploid pool is grown on control and Y2H-selective conditions with 1 mM CuSO$_4$ to induce expression of the AD-Y and DB-X fusions; AD-Y and (Continued on next page)

**Fig 1 (Continued)**

DB-X fused proteins that interact with each other reconstitute the function of the Gal4p transcription factor and drive expression of the reporter gene *GAL1::HIS3*, allowing growth on Y2H-selective medium; (4) after induction of Cre-recombinase expression, the AD-Y downtag and the DB-X uptag recombine with each other in each cell, creating two chimeric barcodes that represent the combined identities of the AD-Y and DB-X vectors in those cells; (5) the chimeric barcodes are PCR amplified using primers containing Illumina adaptor sequences; and (6) Illumina sequencing gives the abundance of each unique chimeric barcode, which reflects the abundance of the cells carrying those specific X–Y combinations in the control or Y2H selection pool. This abundance in turn reflects, in the Y2H selection condition, the interaction between the AD-Y and DB-X fusion partners. (B) Test of the *CUP1* promoter in the AD and DB vectors. Empty AD and DB vectors were grown on medium lacking copper and tryptophan (AD, SC–Trp) or leucine (DB, SC–Leu), backdiluted and grown with and without 1 mM CuSO$_4$. Samples were taken at 0, 3, 6 and 24 h, analyzed on SDS-PAGE and immunoblotted using anti-AD, anti-DB and anti-actin as a loading control. (C) An inducible Y2H assay using known Y2H interaction pairs between yeast growth-inhibitory *L. pneumophila* effectors (underlined) and their antagonist metaeffectors (18) and empty vector controls (EV). Strains were spotted on control (SC–Leu/Trp, selection for presence of both the AD and DB vectors) or Y2H selection conditions: (SC–Leu/Trp/His + 1 mM CuSO$_4$) without (low stringency) or with 1 mM 3-AT (high stringency). The non-toxic DB-fused antagonist metaeffector is mated with the AD empty vector to screen for autoactivation (e.g., the ability to drive *GAL1::HIS3* expression by itself). Plates were incubated for 3 days at 30°C before imaging. SC, synthetic complete.

T4SS, both as positive controls and to potentially discover Dot/Icm protein interactions with effectors. As a positive reference set of human protein interactions (hsPRS), we included 21 well-characterized human protein pairs known to interact (Table S2) (46). Overall, these AD-Y and DB-X fusion collections contain 435 unique ORFs represented by 1,244 uniquely barcoded iBFG-Y2H vectors and 422 unique ORFs represented by 1,137 uniquely barcoded iBFG-Y2H vectors, respectively (Table S1). The majority of ORFs are represented by three unique barcode replicates (Fig. S1A and B). Of the 418 *L. pneumophila* effector and Dot/Icm ORFs, 400 are represented in both AD and DB libraries, whereas the remaining 18 were only screened in one direction due either to high background ("autoactivation") or other technical limitations (Table S3). We mated the AD-Y and DB-X haploid collections *en masse* to generate a pool of AD-DB diploid cells (>1.4 million possible unique barcode combinations) and then induced expression using 1 mM copper during growth on control medium and selective medium lacking histidine. Following growth of the pools, the barcodes of the control condition and the Y2H-selective condition were PCR amplified and sequenced (yielding ~54 and ~72 million read pairs, respectively). Each barcode recombination event results in two chimeric barcode combinations (Fig. 1A, steps 4 and 5). To assess if there are barcode-specific effects due to PCR or sequencing artifacts, we looked at the correlation of the two chimeric barcodes created during AD-Y and DB-X barcode recombination for both pools (Fig. S1C and D), which showed a strong congruence indicating no major barcode-specific effects.

Next, we benchmarked our screen by examining the barcode-fusion data for the 21 expected human–human interactions (Table S2). For each possible X–Y combination in the pool, an interaction score was calculated that reflects the enrichment of the X–Y combination in the Y2H-selective condition over the control condition (Fig. 2A) (46). At the maximum Matthews correlation coefficient (MCC), which optimizes for a balance of high precision and recall, 20 hsPRS interactions exceeded the corresponding interaction score threshold (Fig. 2B; Table S4), a recall of 95%.

Outside of the hsPRS set, 107 Dot/Icm-Dot/Icm, Dot/Icm-effector, effector–effector, and human protein–effector interaction pairs exceeded the MCC-optimal threshold. To verify these, we first recloned each gene and created new individual AD and DB strains. After mating, the resulting diploids were spotted onto control medium and Y2H-selective medium and scored for growth after 3 days. In total, this retesting verified 56 *L. pneumophila* interaction pairs (Fig. 2C; Fig. S2; Table S4), a retest rate of 52%. For perspective, this is comparable to the ~50% verification rate reported for the original BFG-Y2H screen (46) as well as other high-throughput Y2H studies (63–65). Notably, we observed that as the confirmation series approached the MCC-optimal threshold, the Y2H retest-verification rate decreased rapidly, suggesting that the MCC-optimal threshold based on the hsPRS is an appropriate cut-off for *L. pneumophila* interactions (Fig. 2C).

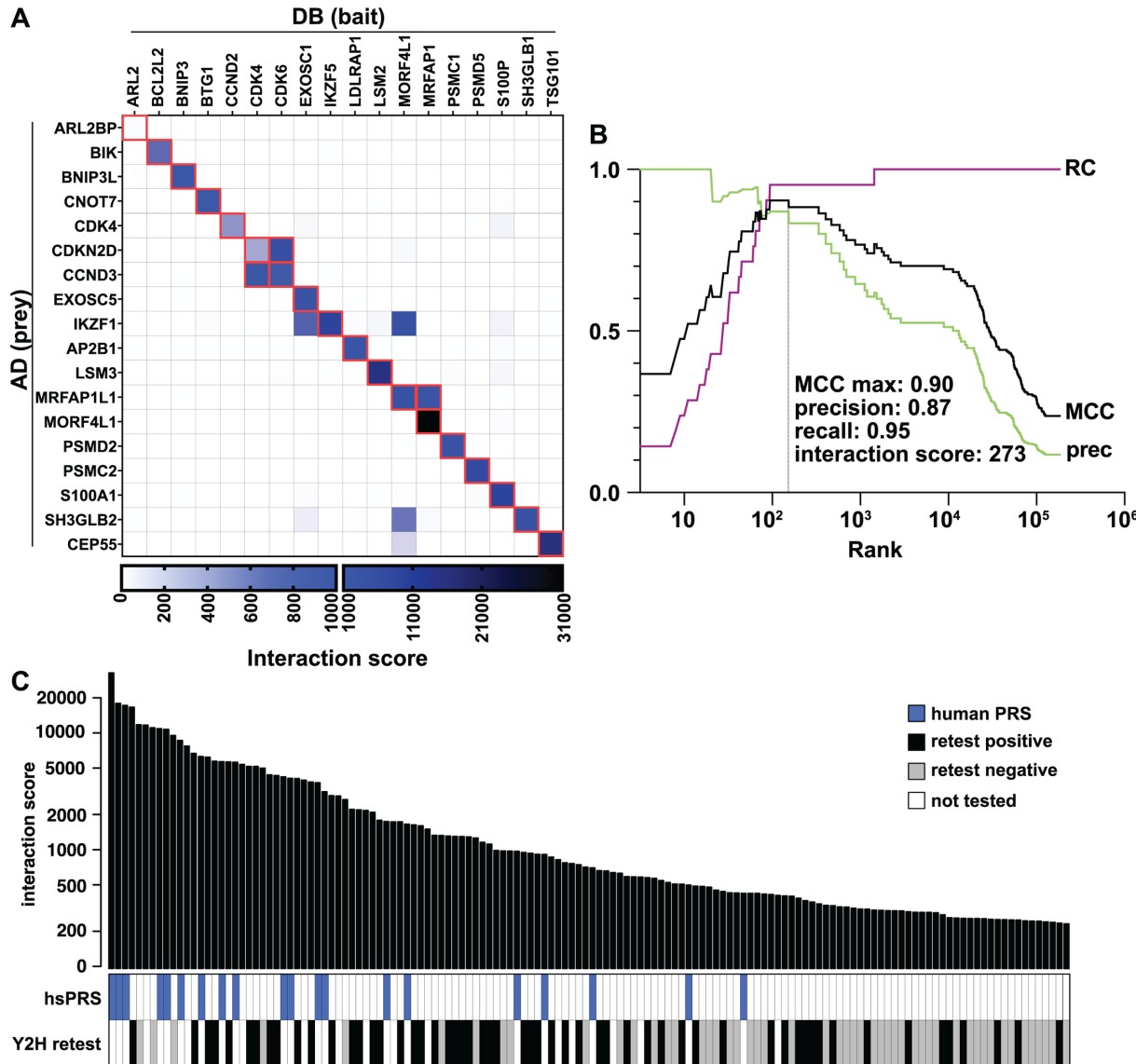

**FIG 2** Performance of the iBFG-Y2H *L. pneumophila* effector screen. (A) Interaction score matrix of all hsPRS ORFs. The 21 expected interaction pairs are highlighted in red. The interaction score heatmap is visualized using two ranges (0–1,000 and 1,000–31,000) to capture the entire score range. (B) MCC (black), prec (green), and RC (purple) plot for the hsPRS. The optimal MCC is indicated with a gray line. The interaction score, precision, and recall values at the optimal MCC are listed. (C) Bar graph showing the ranked interaction scores of the top 140 interaction pairs above the MCC-optimal threshold. The panels below show the location of the human PRS (blue) and the results of the iBFG-Y2H retest experiment (white, not tested; gray, negative; and black, positive). To confirm interaction pairs, ORFs were recloned in inducible AD or DB vectors using confirmed Gateway entry vectors and transformed to the BFG-Y2H yeast strains. Haploid strains carrying the AD or DB vectors were mated, and resulting diploids were selected, spotted on control (SC–Leu/Trp) and Y2H-selective (SC–Leu/Trp/His + 1 mM CuSO$_4$) conditions, and grown for 3 days at 30°C. The images of the retest set are shown in Fig. S2, and the results are listed in Table S4. hsPRS, human positive reference set; MCC, Mathew's correlation coefficient; prec, precision; RC, recall.

Taken together, performance of the iBFG-Y2H screen in our hands matched prior expectations. It captured 95% of the human positive reference set, had a verification rate of 52% for the non-hsPRS interactions, and identified 56 non-hsPRS interaction pairs representing 52 unique PPIs (four interactions were captured in both the AD-DB and DB-AD orientations) (Table S4).

## Validation of iBFG-Y2H interactions by an orthologous (yeast-based NanoLuc two-hybrid) assay

To further evaluate the quality of the *L. pneumophila* iBFG-Y2H interaction data, we next turned to the orthogonal yeast-based NanoLuc two-hybrid (yN2H) assay (66). In this assay, nanoluciferase is split into two fragments: the N-terminal fragment 1 and C-terminal fragment 2. Enzymatic activity of NanoLuc is reconstituted when the two fragments are brought into close proximity by interacting fusion partners. We created copper-inducible NanoLuc two-hybrid (N2H) vectors to assay *L. pneumophila* PPIs in the yN2H assay. We measured the N2H signals of 33 retest-positive *L. pneumophila* interaction pairs as N1-X:N2-Y and N1-Y:N2-X fusions along with a set of well-described interactions (positive reference set, hsPRS-v2) and randomly selected pairs (random reference set, hsRRS-v2) (66–69) cloned into the standard N2H vector (66) (Table S5). The detection rate for the *L. pneumophila* interactions pairs in a single orientation was compared to the detection rate of the hsPRS-v2 and hsRRS-v2 (66) (Fig. 3A). The detection threshold was set at the normalized luminescence ratio (NLR) value, where the hsRRS-v2 distribution has a $Z$-score of >2.23 and the probability of the hsRRS-v2 values below the threshold is 98.7%. The N2H detection rate of our data set was 48.5%, which exceeds the hsPRS-v2 set detection rate of 18.3% (Fig. 3B), so that we can consider our entire interaction data set to be well validated. Indeed, by this measure, the quality of the verified *L. pneumophila* iBFG-Y2H interactome exceeds that of a high-quality interaction data set supported by multiple experiments in the curated literature (66–70).

## The iBFG-Y2H screen identifies both known and novel Dot/Icm complex interactions

Returning to the biology of *L. pneumophila* interactions, we first looked at components of the Dot/Icm T4SS. The interaction score heatmap of the 27 DB- and 28 AD-fused Dot/Icm components present in the screen shows nine interaction pairs above the MCC-optimal threshold and verified in the iBFG-Y2H retest assay (Fig. 4A; Fig. S2; Table S4). We expected to see three interactions: IcmQ–IcmR (71), IcmS–IcmW (72–74), and LvgA–IcmS (75, 76), which have all been detected using several PPI assays, including the Y2H assay. IcmQ–IcmR and IcmS–IcmW were captured in both directions (Fig. 4B)

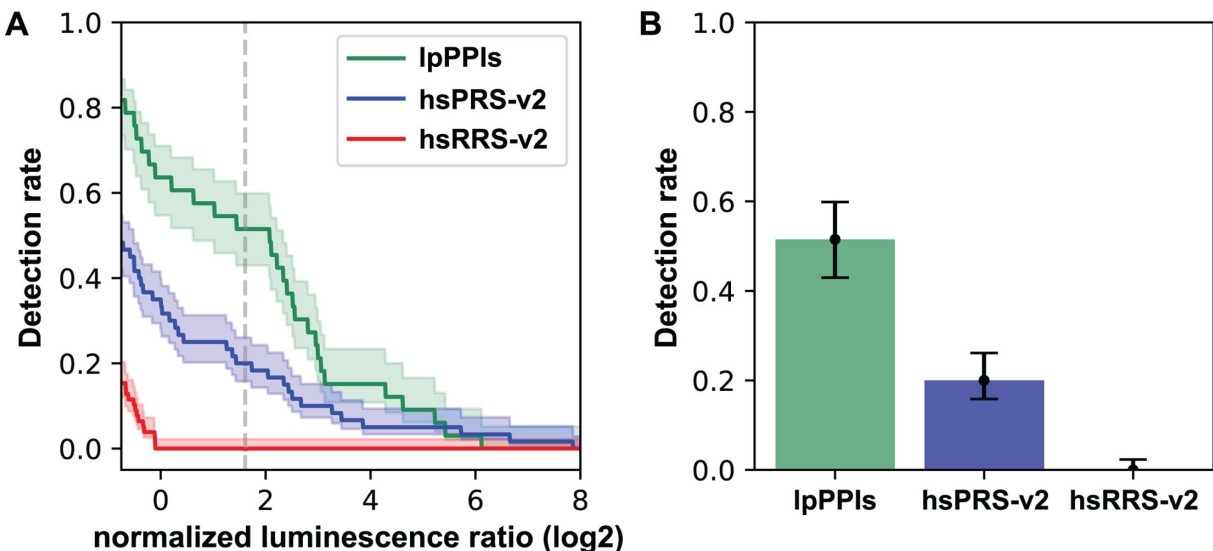

**FIG 3** Validation of the iBFG-Y2H interaction pairs by the orthologous yN2H method (A) The log-transformed NLR from the yN2H experiment is plotted for a random sampling of the human PRS (hsPRS-v2), RRS (hsRRS-v2), and 33 iBFG-Y2H pairs that were verified in the iBFG-Y2H retest screen (IpPPIs). The dashed line indicates log2 NLR value of 1.776, where the $Z$-score for the RRS is 2.23. Confidence clouds represent a 68.3% Bayesian confidence interval. (B) The detection rate of the IpPPI, hsPRS-V2, and hsRRS-V2 based on the threshold in panel A. Error bars represent a 68.3% Bayesian confidence interval. The detection rate of the *L. pneumophila* pairs exceeds the detection rate of the hsPRS-V2 pairs, suggesting robustness of this interaction set. NLR, normalized luminescence ratio.

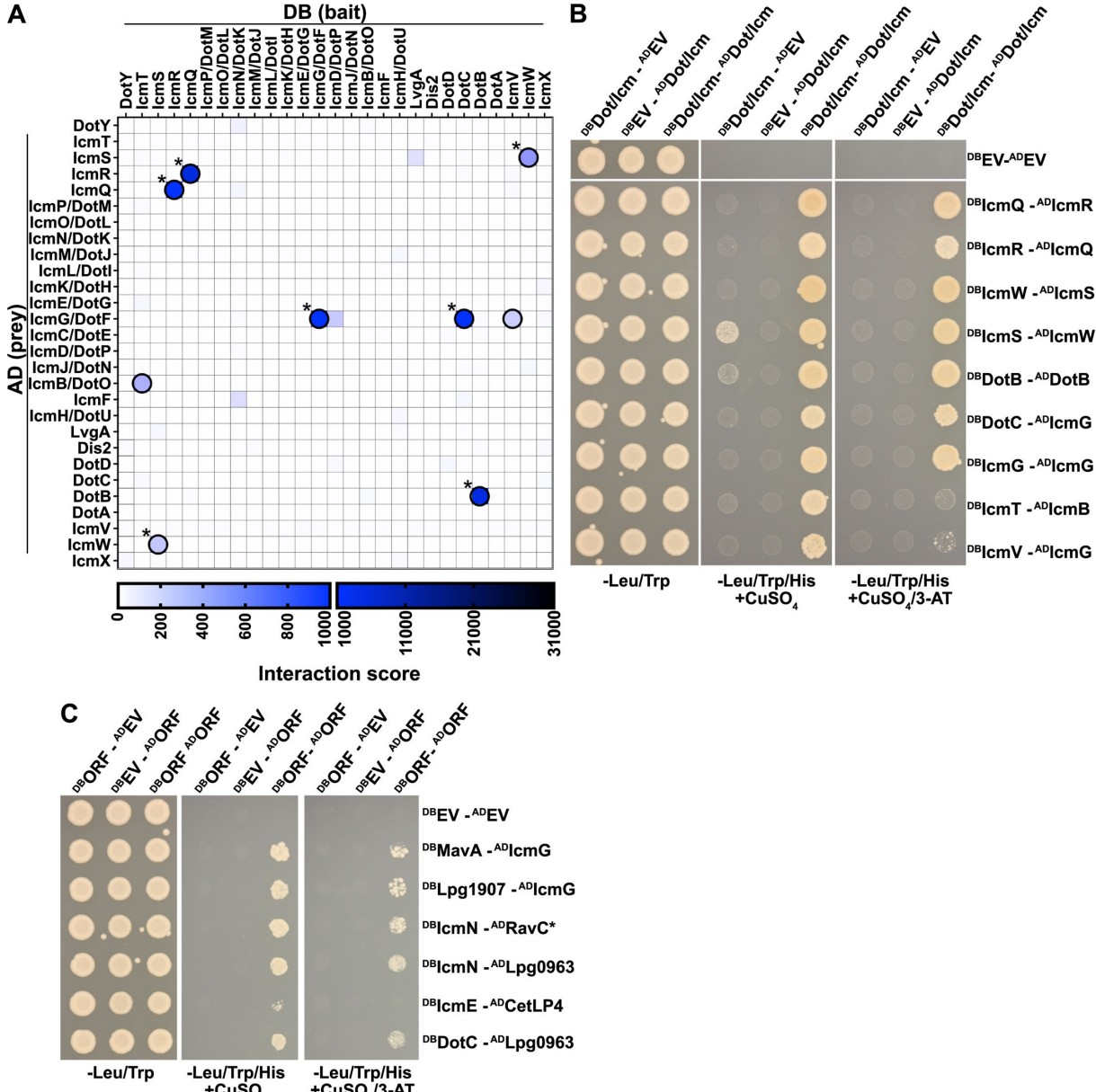

**FIG 4** iBFG-Y2H captures seven interactions of the Dot/Icm complex. (A) The Dot/Icm T4SS interaction score matrix showing the 27 DB- or 28 AD-fusions of the Dot/Icm components present in the screen. The nine interaction pairs that were positive in the iBFG-Y2H retest screen (Fig. S2) are circled in black; published interactions are indicated with a star. (B) Inducible Y2H assay for nine verified Dot/Icm-Dot/Icm interaction pairs on different Y2H-selective conditions. X–Y pairs that were positive in the retest screen (Fig. S2) were grown on diploid-selective medium (−Leu/Trp) and two Y2H-selective conditions: the low stringency condition used in the retest screen (−Leu/Trp/His + 1 mM CuSO4) and a higher stringency condition (−Leu/Trp/His + 1 mM CuSO4 + 1 mM 3-AT). To assay for autoactivator activity of the DB-X or AD-Y fusion, each DB-X or AD-Y fusion is mated with AD or DB empty vector, respectively. DB-IcmS is an autoactivator (the ability to grow on −Leu/Trp/His + 1 mM CuSO₄ in the absence of an DB-AD complex). In the higher stringency condition (−Leu/Trp/His + 1 mM CuSO₄/1 mM 3-AT), the DB-IcmS–AD-IcmW diploid can still grow, but DB-IcmS with AD-empty vector diploid cannot. (C) Inducible Y2H assay of verified effector–Dot/Icm interaction pairs on different Y2H-selective conditions, as described above. The core effector RavC, conserved across *Legionella* spp., is indicated with a star.

above the MCC-optimal threshold and verified in the iBFG-Y2H retest assay, but the LvgA–IcmS pairs were below the MCC-optimal threshold and were thus not further tested or assessed by yN2H. Of the five additional verified PPIs from the iBFG-Y2H assay, three are supported by previously published data: DotB-DotB, IcmG/DotF-IcmG/DotF, and DotC-IcmG/DotF (Fig. 4B). The ATPase DotB can be purified as hexamer (77); IcmG/DotF was shown to self-associate using the bacterial adenylate cyclase two-hybrid

(BACTH) assay (78) and DotC and IcmG/DotF are both part of the T4SS core complex (78). The remaining two PPIs, IcmB/DotO-IcmT and IcmV-IcmG/DotF, are novel.

In contrast to a previous Y2H study identifying interactors of IcmW (73), our assays did not identify any effector interactions with any of the components responsible for transferring effectors to the core transmembrane complex. This transfer is the function of the Type IV coupling complex (T4CC) of which seven components are present in the iBFG-Y2H library (IcmO/DotL, IcmP/DotM, IcmJ/DotN, DotY, IcmS, IcmW and LvgA) (74, 79–83). However, we did find interactions of effectors with Dot/Icm components that are part of the core complex spanning the inner membrane, periplasm and outer membrane: IcmG/DotF, IcmN/DotK, IcmE/DotG and DotC (Fig. 4C; Fig. S2; Table S4) (78, 84, 85). Notably, IcmG/DotF was previously shown to interact with effectors in BACTH assays (86, 87), though the relevance of these interactions is unclear (87).

Taken together, we captured five known interactions from the Dot/Icm T4SS; uncovered two novel Dot/Icm interactions, IcmB/DotO-IcmT and IcmV-IcmG/DotF; and six effector-Dot/Icm T4SS interactions.

## iBFG-Y2H identifies 10 novel effector homodimers and 10 novel effector–effector interactions

Next, we looked at effector–effector interactions verified in the iBFG-Y2H retest assay (Fig. 5; Fig. S2; Table S4). The screen captured 11 effector homodimers (Fig. 5A), of which only WipA had previously been shown to dimerize (88). Beyond this, we identified 13 physical interactions between pairs of distinct effectors (Fig. 5B). These include the published effector–metaeffector pairs RavJ-LegL1 (in both directions) and LegC7/YlfA-MavE (18), consistent with the results of our pilot experiment (Fig. 1C). Ten PPIs are novel effector–effector pairs. Of these, three contain a core (conserved) effector—RavC, CetLP1, or Lpg2832 (89, 90)—and three have at least one effector with some

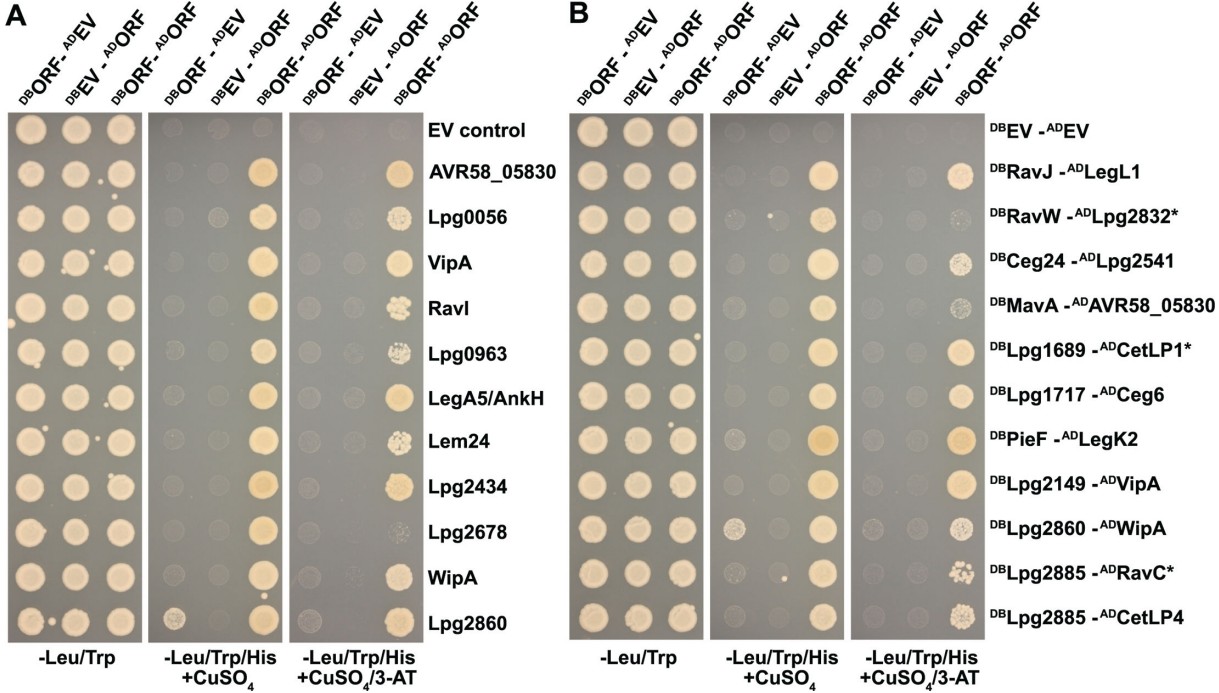

**FIG 5** iBFG-Y2H effector–effector protein interactions. (A) Inducible Y2H assay of verified interactions of effectors and putative effectors that self-interact on different Y2H-selective conditions. The 11 X–Y pairs that tested positive in the retest screen (Fig. S2), and their AD or DB empty vector control combinations were grown on diploid-selective medium (−Leu/Trp) and two Y2H-selective conditions: the low stringency condition used in the retest screen (−Leu/Trp/His + 1 mM CuSO₄) and a higher stringency condition (−Leu/Trp/His + 1 mM CuSO₄/1 mM 3-AT). (A and B) DB-Lpg2860 is an autoactivator in the low stringency condition but not in the higher stringency condition. (B) Inducible Y2H assay of 11 AD-DB novel pairs that were positive in the retest screen and involve two effectors or putative effectors on different Y2H-selective conditions, as described above. Core effectors conserved in all *Legionella* spp. are indicated with a star.

characterization: Lpg2149-VipA (33, 91, 92), PieF-LegK2 (93–95), and WipA-Lpg2860 (60, 88, 96). The remaining four pairs identified in our screen consist of completely uncharacterized effectors and putative effectors.

## An incidental consequence of the human positive reference set: identification of 17 effector–host interactions

While our screen focused on effector–effector interactions, one consequence of using a set of evolutionary conserved human proteins to benchmark a pooled interaction screen was the fortuitous scoring of interactions between effectors and these conserved host proteins. Eight proteins of the hsPRS, which contains conserved eukaryotic proteins such as proteins involved in cell cycle regulation or mRNA degradation, were also found to interact with *L. pneumophila* effectors. We identified 17 effector–human protein pairs (Fig. 6; Fig. S2; Table S4): 8 effectors interact with transcription factor Ikaros (IKZF1) (Fig. 6A), and the remaining 9 pairs involve 6 different effectors and 7 human proteins (Fig. 6B). Of these interactions, 16 are novel, and only the PieF-CNOT7 interaction was reported previously (95).

## A high-confidence effector interaction network

Using iBFG-Y2H, we identified 2 novel Dot/Icm-Dot/Icm interactions, 6 novel effector-Dot/Icm interactions, 10 novel effector dimers, 10 novel effector–effector interactions, and 16 novel effector–human protein interactions. These PPIs and previously published PPIs captured in the iBFG-Y2H screen are visualized in a network (Fig. 7), where nodes of published interactions are colored blue. Examination of this network shows that while most effectors only interact with one other effector or human protein, a few effectors stand out as interacting with several different proteins: PieF (interacting with IKFZ1, CNOT7, CDK4, LSM3, and LegK2), VipA (with CDK4 and Lpg2149), Lpg2885 (with Lpg1822 and CetLP1), and MavA (with IKZF1, MRFAP1L1, and AVR58-05830).

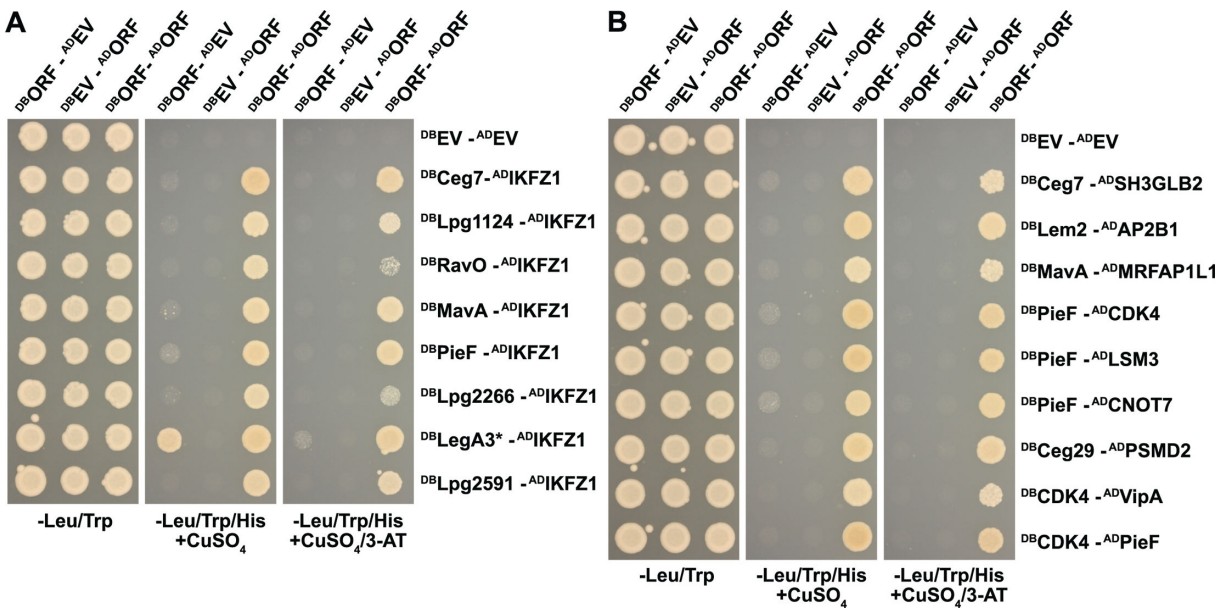

**FIG 6**  iBFG-Y2H interactions of human positive reference set proteins with effectors and putative effectors. (A) Inducible Y2H assay of verified interactions of effectors and putative effector Lpg2266 with human transcription factor IKZF1 on different Y2H-selective conditions. The eight X–Y pairs that tested positive in the retest screen (Fig. S2) and their AD or DB empty vector control combinations were grown on diploid-selective medium (−Leu/Trp) and two Y2H-selective conditions: the low stringency condition used in the retest screen (−Leu/Trp/His + 1 mM CuSO$_4$) and a higher stringency condition (−Leu/Trp/His + 1 mM CuSO$_4$/1 mM 3-AT). The core effector LegA3 (indicated with a star) is an autoactivator on low stringency medium but not in the higher stringency condition. (B) Inducible Y2H assay of nine verified interaction pairs (Fig. S2) between effectors and eight proteins from the human positive reference set on different Y2H-selective conditions, as described above.

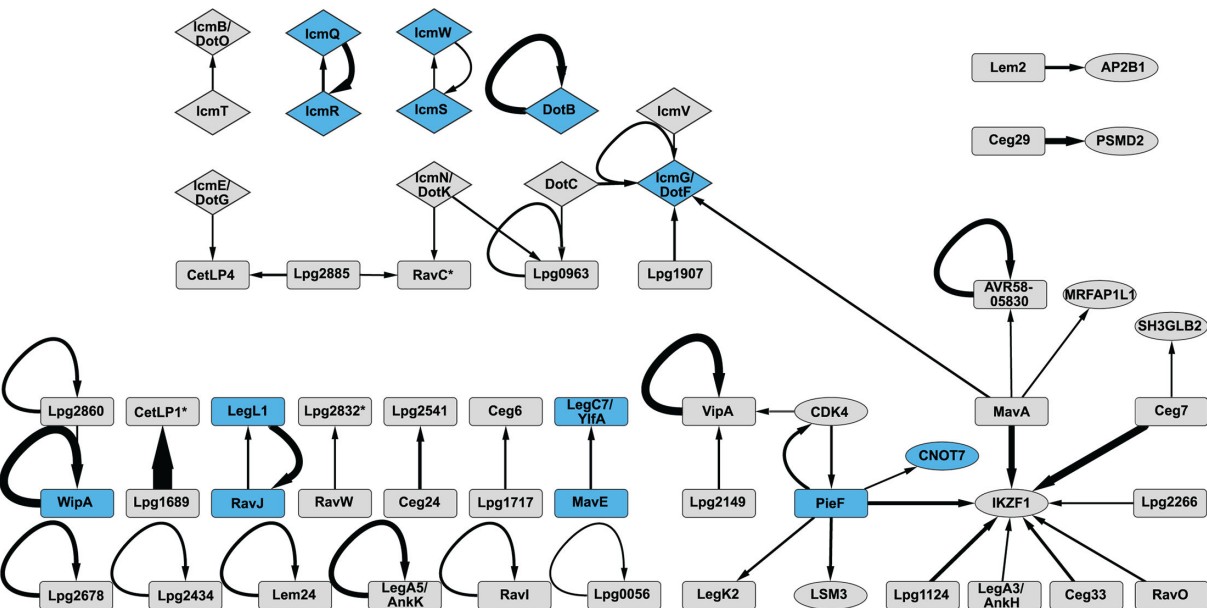

**FIG 7** A network view of the iBFG-Y2H interaction pairs. A network showing all the verified interaction pairs captured in the iBFG-Y2H screen involving effectors or putative effectors. Effectors are shown as rectangles, Dot/Icm components as a diamond, and human PRS proteins as an oval. Arrows point from DB to AD; the thickness of the edges is reflective of the iBFG-Y2H interaction score. Core effectors are indicated by a star and nodes of previously published PPIs are shown in blue. The network was created using Cytoscape v.3.101 (97).

## DISCUSSION

With over 300 effectors, *L. pneumophila* has the largest described bacterial effector arsenal (98) with several observed instances of effector interplay that fine-tune effector function and the progression of pathogenesis (5–9, 14, 16, 18, 33–36, 38, 39). We previously performed the first genetic interaction screen of bacterial effectors, where we expressed every possible combination of *L. pneumophila* str. *Philadelphia 1* effectors in the budding yeast *S. cerevisiae* and identified several suppression pairs, where an antagonist effector suppressed the yeast-growth defect caused by a growth-inhibitory effector (18). These suppression pairs were enriched for metaeffectors, effectors that directly target other effectors and regulate their activity in the host cell. Collectively, the field has identified 11 *L. pneumophila* metaeffectors to date (16, 18, 33, 39). We reasoned that our genetic interaction screen did not capture all metaeffectors or effectors otherwise functioning in a physical complex (15). To complement our previous work, we set out to screen all possible effector–effector physical interactions using iBFG-Y2H, a high-throughput, multiplexed protein interaction screen with inducible expression and barcoded vectors for a sequencing readout.

In this study, we present the systematic physical interaction screen of a large bacterial pathogen effector arsenal, encompassing 390 *L. pneumophila* effectors and putative effectors and 28 Dot/Icm components. We identified 52 interactions between *L. pneumophila* effectors, Dot/Icm T4SS components, and effectors with Dot/Icm T4SS components or human proteins. A subset iBFG-Y2H PPIs were detected in the orthologous N2H assay (66) at a rate of 48.5% (Fig. 3). This is better than the detection date for a high-quality positive reference set, indicating that the verified iBFG-Y2H set is a high-confidence data set, with 16 PPIs captured in both assays of which 9 are novel (see Table S5). Our screen captured five known interactions between components of the Dot/Icm T4SS system (71–74, 76–78): the WipA dimer (88), two of our previously published effector–metaeffector pairs (RavJ-LegL1 and LegC7/YlfA-MavE) (18), and PieF-CNOT7 (95) (Fig. 7). The 10 novel effector–effector interactions involve 19 effectors, approximately 5% of the total effectors, and doubles the number of known PPIs between *L. pneumo-*

*phila* effectors. This reinforces the notion that bacterial effectors do not act on their own and need to be studied in concert rather than in isolation.

The functional consequences of the effector–effector interactions we identified remain to be defined. They may represent metaeffector–effector pairs, effectors functioning in a complex, or some other functional relationship yet to be discovered. Nevertheless, these physical interactions already suggest interesting functional links. For example, Lpg2149 links the actin nucleator VipA (91) with modulators of the E3 ubiquitin ligase UBE2N, MavC and MvcA. As previously reported, MavC modifies and inactivates UBE2N (33, 34, 36) and MvcA removes that modification (36, 99). Lpg2149 is the metaeffector of MavC and MvcA, and inactivates both effectors (33). Notably, UBE2N has been shown to be involved in ubiquitination of β-actin (100) and is recruited to actin-rich structures in *Listeria monocytogenes* infections (101).

A fortuitous consequence of including the human positive reference set in our screen was the detection of 16 novel host protein–effector interactions. The transcription factor Ikaros (IKZF1) stands out, as it interacts with a striking number of effectors. Ikaros is involved in regulation of the host response to bacterial lipopolysaccharide (LPS) (102) and hypomorphic Ikaros alleles leave patients at high risk of viral and bacterial infections (103). This raises the question of whether one or more of these effectors modulate Ikaros-regulated host defenses. Indeed, *L. pneumophila* has been shown to target other immune-related transcription factors such as nuclear factor-κB (NF-κB) (62, 104). We also identified an effector, Ceg29, that may be linked to the tumor necrosis factor (TNF) signaling pathway known to restrict *L. pneumophila* growth (105, 106): Ceg29 interacts with the proteasome non-ATPase regulatory subunit 2, PSMD2, which can bind to TNF receptor and is implicated in TNF signaling (107, 108).

Two other human protein–effector PPIs that can be linked to known processes during *L. pneumophila* infection are (i) AP2B1-Lem2 and (ii) SH3GLB2 (Endophillin B2)-Ceg7. AP2B1 is a component of the clathrin adaptor complex, AP2, and is involved in endocytosis, while endophillin B2 facilitates endosome maturation. Both processes are heavily targeted by *L. pneumophila* as it evades host defenses and maturation of the host phagosome (109). Interestingly, the AP2 complex is also targeted by an effector of *Coxiella burnetii* (110), a related bacterial pathogen with an intracellular lifestyle. Finally, during infection, *L. pneumophila* blocks the host cell cycle progression in an effector-dependent manner (111, 112). Two effectors, VipA and PieF, were found to interact with the cyclin-dependent kinase, CDK4, suggesting that they could be involved in cell cycle regulation.

Beyond the positive hits of the iBFG-Y2H screen, a number of known interactions were not captured by the iBFG-Y2H screen. These include the metaeffector LubX and its effector target, SidH, as well as metaeffector–effector pairs that we previously identified in targeted Y2H interrogation: SidP–MavQ, LupA-LegC3, and SdbC-SdbB (18). There are likely several reasons for this, including known sensitivity limitations (e.g., not optimized for membrane protein interactions, interactions depending on specific subcellular localization or post-translational modifications) (113), the confounding effects of yeast toxicity despite our inducible plasmids (18, 39), and protein instability (e.g., SidH is polyubiquitinated by LubX and subject to proteosomal degradation) (16). This rate of "false negatives" is not unexpected as it is well established that any one binary PPI assay is likely to detect only approximately one-third of high-confidence, benchmarked interactions (66, 67, 70, 114–119). Altogether, this cautions against interpreting lack of detection as either a lack of interaction or a unique limitation of the assay as performed. Instead, it suggests that more effector–effector interactions are likely to be revealed through the application of several complementary, orthologous methodologies (66).

In summary, our iBFG-Y2H screen of all *L. pneumophila* effectors captured a novel set of PPIs, which builds on and expands our knowledge of the molecular interactions between *L. pneumophila* effectors that facilitate *L. pneumophila* pathogenesis. Our data set doubles the number of known effector–effector interactions and shows that effector–effector interactions and effector interplay are common, rather than an

exception. We herein present this data set as a resource to the field. The next steps in studying these interactions, such as investigating the role of these interactions during host infection, will undoubtedly lead to novel biology and a greater understanding of regulation of pathogenesis of intracellular bacterial pathogens.

## MATERIALS AND METHODS

### Strains and culture conditions

*Escherichia coli* strain Top10 was used for cloning and plasmid production and grown in LB Miller or 2× LB Miller. *S. cerevisiae* strains RY1010 (MATa *leu2-3,112 trp1-901 his3-200 ura3-52 gal4Δ gal80Δ PGAL2-ADE2 LYS2::PGAL1-HIS3 MET2::PGAL7-lacZ cyh2$^R$ can1Δ::PCMV-rtTA-KanMX4*), RY1030 (*MATα leu2-3,112 trp1-901 his3-200 ura3-52 gal4Δ gal80Δ PGAL2-ADE2 LYS2::PGAL1-HIS3 MET2::PGAL7-lacZ cyh2$^R$ can1Δ::TADH1-PtetO2-Cre-TCYC1-KanMX4*) (46), Y8800 (MATa), and Y8930 (MATα) (genotype: *leu2-3,112 trp1-901 his3-200 ura3-52 gal4Δ gal80Δ PGAL2-ADE2 LYS2::PGAL1-HIS3 MET2::PGAL7-lacZ cyh2$^R$*) were grown in yeast extract, peptone, adenine dextrose (YPAD) medium (2% bacto peptone wt/vol, 1% yeast extract wt/vol, 2% glucose vol/vol, 180 mg/L adenine) or synthetic complete (SC) medium lacking specific amino acids with 2% glucose and 180 mg/L adenine. The SC medium for copper-inducible strains was prepared using yeast nitrogen base (YNB) without amino acids or copper (ForMedium, catalog # CYN0905).

### Inducible BFG-Y2H vectors

The constitutive *ADH1* promoter in the BFG-Y2H vectors pNZM1090 and pNZM1100 (46) was replaced with the *CUP1* promoter sequence (61). First, a second HindIII site in pNZM1090 was removed by introducing synonymous substitutions (in the *TRP1* gene) with QuikChange (Agilent) per manufacturer's instructions using primers pNZM1090F and pNZM1090R (Table S6), resulting in pNZM1090-HindIII. The *CUP1* promoter was amplified from *S. cerevisiae* BY4741 genomic DNA (120) using primers AE897 and AE898. The resulting PCR product was digested with ApaI/HindIII and ligated into ApaI/HindIII digested pNZM1090-HindIII and pNZM1100. The resulting vectors pNZM1090CUP1 and pNZM1100CUP1 were Sanger sequence verified.

Inducible expression was tested by growing RY1010 pNZM1090CUP1 and RY1030 pNZM1100 strains on SC medium lacking tryptophan or leucine, respectively. Overnight cultures were diluted to 1 $OD_{600\,nm}$/mL and induced with or without 1 mM $CuSO_4$. Three $OD_{600\,nm}$ units were harvested at 0, 3, 6, and 24 h. Samples were lysed as described previously (121) and resuspended in 100 µL 2× sample buffer (4% SDS, 20% glycerol, 120 mM Tris, pH 6.8). The equivalent of 0.3 OD units was analyzed by SDS-PAGE and Western blot using the following antibodies and dilutions: mouse anti-AD (Abcam, catalog # ab135398) 1:200 in 3% bovine serum albumin (BSA) in phosphate-buffered saline–0.1% Tween (PBS-T), mouse anti-DB (Abcam, catalog # ab135397) 1:1,000 in 3% BSA in PBS-T, rabbit anti-actin (MilliporeSigma, catalog # A2066) 1:2,500 in 5% milk PBS-T, and secondary antibodies anti-mouse horseradish peroxidase (HRP) (1:5,000 for anti-AD and 1:10,000 for anti-DB in 5% milk PBS-T) and anti-rabbit HRP (1:10,000 in 5% milk PBS-T) (Cell Signaling Technology, catalog # 7076 and 7074).

### Barcoded iBFG-Y2H plasmid collection

Randomly barcoded iBFG-Y2H vectors were made as described previously (46) with the modification that the barcode cassette was inserted at the SacI site downstream of the Gateway cassette.

A Gateway pDONR221 library containing confirmed and putative *L. pneumophila* effectors described previously (18, 62) was a kind gift from Dr. Ralph Isberg. We cloned an additional 52 effectors and putative effectors and 26 Dot/Icm components. ORFs were amplified from genomic DNA from *L. pneumophila* strain Lp02 (122) using the primers listed in Table S6. The PCR products were cloned into pDONR221-ccdB (Invitrogen) using

Gateway BP clonase II (Invitrogen) per manufacturer's instructions, and the resulting vectors were sequence verified. The pDONR221 library was cloned *en masse* using Gateway LR Clonase II (Invitrogen) into randomly barcoded iBFG-Y2H vectors. A set of 34 human ORFs (Table S2) for calculation of precision, recall, and MCC values was included as described (46).

The randomly barcoded pools were transformed into chemically competent Top10 *E. coli* cells, and transformants were arrayed in a 384-well format using an S&P robotic rearrayer (S&P Robotics). Barcode and ORF sequences for each clone were determined by kiloSEQ (seqWell Inc). Up to three independent clones (with unique barcodes) were chosen for each ORF. Missing ORFs were cloned individually into randomly barcoded iBFG-Y2H vector pools and transformed into Top10 *E. coli*, and six clones were selected and arrayed into 96-well plates. The barcodes were sequenced using pooled Illumina sequencing with row–column–plate barcodes to link vector barcodes to a unique plate and well identity (46). Briefly, *E. coli* cultures were grown overnight in LB medium with 100 µg/mL carbenicillin at 37°C with shaking in 96-well plates and diluted 1/20 in ddH$_2$O. Ten-microliter row–column PCR reactions were performed with 1 µL barcoded primer corresponding to row A–H and 1 µL barcoded primer for column 1–12 barcoded primer (oHM106-199) (2 µM stock), 5 µL of KAPA HiFi 2× master mix (Roche), and 3 µL of diluted overnight culture grown in 96-well plates. Following PCR amplification, the amplicons from a 96-well plate were pooled and purified using a 2% E-Gel SizeSelect II agarose gel (Invitrogen) and quantified on a NanoDrop spectrophotometer (Thermo Fisher Scientific). Purified amplicon pools then underwent a second PCR amplification adding Illumina flow cell adapters as well as additional inline plate barcode sequences (primers oHM146-147 and oHM200-213). For each plate, a 40 µL KAPA HiFi reaction was assembled using 1 µL of each primer (oHM200-213, 10 µM stock) and 5 µL of amplicons (1 ng/µL) and purified as above. The purified products were quantified using the NEBNext Library Quant kit for Illumina (NEB) and sequenced using a mid-output reagent cartridge with 2 × 150 paired-end reads on an Illumina Miniseq platform. Reads were demultiplexed using a custom Perl script (46) and aligned to a vector sequence using BLAST+ (123) to extract barcode identities. Up to three independent clones were selected for each ORF. Plasmid pools were purified from the final arrayed *E. coli* collections using PureYield Plasmid Midipreps (Promega).

## iBFG-Y2H screen

The iBFG-Y2H screen was performed using the *S. cerevisiae* BFG-Y2H toolkit strains RY1010 and RY1030 strains (46). Frozen competent yeast cells were prepared as described (124) and transformed as follows: 2 mL of frozen competent RY1010 or RY1030 cells was thawed, pelleted, and resuspended in an 8.280 mL yeast transformation mixture (33% PEG3350, 0.1 M lithium acetate, and 0.3 mg/mL boiled salmon sperm DNA) with 40 µg of the AD or DB iBFG-Y2H vector pool and heat-shocked at 42°C for 1 h. The cells were pelleted at 1000 × *g* for 5 min, washed once with 10 mL ddH$_2$O, resuspended in ~1 mL ddH$_2$O, and plated on eight 15 cm plates of SC medium lacking tryptophan (−Trp for AD vectors), or lacking leucine (−Leu for DB vectors) using YNB without copper. Transformants were grown at 30°C for 3 days, before being scraped from the plates with ddH$_2$O and pooled. AD and DB pools were pelleted, washed twice with 25 mL of ddH$_2$O, and resuspended to 100 OD$_{600 nm}$ units/mL. To mate, equal volumes (30 mL) of the AD and DB pools were combined and incubated for 3 h at room temperature (RT) without shaking (125). The cells were pelleted, resuspended in 1.5 mL of ddH$_2$O, and plated on 10 YPAD plates of 15 cm and incubated at RT for 3 days. The mated yeast pool was scraped, pelleted, and washed twice and resuspended in ddH$_2$O to 50 OD$_{600 nm}$/mL. The mated pool was plated on 18 plates of 15 cm (200 µL per plate) of Y2H-selective medium (SC–Leu/Trp/His + 1 mM CuSO$_4$) or diploid-selective medium (SC–Leu/Trp) supplemented with 1 mM CuSO$_4$ and 8 mM excess of histidine and incubated at 30°C for 3 days. For each condition, the plates were then scraped, pooled, washed twice with ddH$_2$O, and diluted to 1 OD$_{600 nm}$/mL in 100 mL of diploid-selective media without CuSO$_4$ and

with 10 µg/mL of doxycycline to induce Cre-recombinase expression. To allow *in vivo* Cre-mediated recombination of barcodes, the culture was grown overnight at 30°C with shaking at 200 rpm until the $OD_{600\ nm}$ exceeded 5. Plasmids pools were isolated using the Zymoprep II yeast plasmid miniprep kit (Zymo Research). The fused DNA barcode sequences were amplified using KAPA HiFi 2× master mix and fusion-specific primer pairs for each treatment (oHM380-387). For each condition, 20 reactions of 40 µL were run with 2 ng of DNA template per reaction to reduce sampling error and pooled together. The primers were used at 10 µM and contained adapters for the Illumina flow cell as described (46). The PCR amplicon pools were then purified using 0.7× AMPure magnetic beads (Beckman Coulter) following the Illumina Nextera XT recommendations. The concentration of the purified amplicons was quantified by quantitative PCR using the NEBNext Illumina Quant kit (NEB). The forward and reverse reads were demultiplexed, and fused barcodes were quantified through alignment against custom barcode and primer sequences using Bowtie2 (v.2.3.4.1) (-q –local –very-sensitive-local -t -p 23 -reorder) (126).

## Interaction score calculation

Interaction scores were calculated as described previously (46). Briefly, (i) a constant value of 1 was added to every AD-DB barcode combination in both the selective (−Leu/Trp/His + 1 mM $CuSO_4$) and non-selective (−Leu/Trp + 1 mM $CuSO_4$/8 mM His) matrices; (ii) the marginal frequency of each AD or DB clone within the population was determined by dividing the sum of barcode counts for all clones that contain that barcode in the non-selective condition by the sum of all barcode counts in the non-selective matrix. The expected frequency of any AD-DB combination is the product of each clone's marginal frequency in the non-selective condition; (iii) to score enrichment in the selective condition, the frequency of each AD-DB combination was calculated by the dividing barcode count for every AD-DB combination by the sum of all barcode counts in the selective matrix; raw score values ($S$) were then determined by dividing the selective frequency by the expected marginal frequency product determined in (ii); (iv) autoactivation was normalized across each DB clone. First, the median value of all $S$ values for each DB clone was subtracted from each raw score ($S$) giving a new value ($S°$). $S°$ was then divided by the $S$ value that encompasses 60% of all interactions for that DB clone, resulting in the interaction score $S'$. For each AD-DB pair, multiple $S'$ scores were calculated based on the number of barcodes for each clone in the pool and the two chimeric barcodes for each AD-DB combination. Through systematic determination, we found that the optimal Matthews correlation coefficient (maximizing precision and recall), where MCC optimal = 0.9, was achieved with a threshold where the top 60% of interactions were included in the normalization ($\rho = 0.4$) and the average of the top 8 $S'$ signals was adopted.

## Pairwise retesting of iBFG-Y2H candidate interactions

Of the 140 PPI pairs above the MCC-optimal rank, we retested 107 pairs involving Dot/Icm-Dot/Icm pairs, effector-Dot/Icm pairs, effector–effector pairs, and effector–human protein pairs. Validated Gateway entry clones were recloned into unbarcoded pNZM1090CUP1 and pNZM1100CUP1 vectors using Gateway LR Clonase II (Life Technologies) per manufacturer's instructions, transformed to RY1010 and RY1030 as described (127) and grown for 2 days at 30°C on SC–Trp or SC–Leu agar plates. The resulting haploid strains were arrayed in a 96-well format as a DB (RY1030 pNZM1100CUP1) and AD (RY1010 pNZM1090CUP1) array. Using an S&P pinning robot (S&P robotics), the DB and AD arrays were pinned together on YPAD agar plates and incubated overnight at 30°C. The mated, diploid strains carrying both plasmids were then pinned onto SC–Leu/Trp agar plates and grown for 2 days at 30°C. An empty vector control was included on each array plate. To check for autoactivator activity, the DB and AD arrays were mated to an AD or DB empty vector control strain, respectively. The resulting diploid plates were grown overnight in 100 µL SC–Leu/Trp medium in 96-well

plates at 30°C, diluted 10-fold in fresh medium, and spotted on SC–Leu/Trp (control), SC–Leu/Trp/His + 1 mM $CuSO_4$ (Y2H-selective condition) agar plates using a 96-well pin tool (V&P404, V&P Scientific) and grown for 3 days at 30°C before imaging (Fig. S2; Table S4). The retest-positive pairs were subsequently tested on two Y2H-selective conditions: SC–Leu/Trp/His+ $CuSO_4$ and the more stringent condition with 1 mM 3-AT to assess the strength of the interactions and to assay clones with autoactivator activity.

## Pairwise validation of *L. pneumophila* effector interactions by yN2H assay

The promoter sequence of the N2H (N1 and N2) vectors (66) was replaced with the *CUP1* promoter. The *CUP1* promoter sequence was amplified from the iBFG-Y2H vectors (primers oHM487/488) and cloned into iN2H N1 and N2 vectors digested with SpeI/SacI using NEBuilder assembly (NEB) according to manufacturer's instructions. The resulting vectors, pHM526 and pHM527, were Sanger sequence verified.

ORFs from the iBFG-Y2H-verified set were cloned into iN2H pDEST vectors using LR Clonase II as described above. Bacterial transformants were grown overnight in LB + carbenicillin (100 µg/mL); 200 µL of culture was pelleted and resuspended in 130 µL. Fifteen microliters of cell suspension was incubated for 30 min at 27°C with 15 µL of 2× bacterial lysis buffer (2 mg/mL lysozyme, 20 mM Tris–HCl, pH 6.8, and 2 mM EDTA). The lysates were incubated for 20 min at 55°C with 5 µL of proteinase K mixture (12 mg/mL proteinase K, 20 mM Tris–HCl, 2 mM EDTA) followed by 20 min at 80°C to inactivate proteinase K. Competent yeast cells (Y800 and Y8930) were prepared as described; (127) 20 µL of competent Y8800 or Y8930 cells was pelleted, resuspended in 148 µL of yeast transformation mix, and combined with 15 µL of lysate for transformation. Transformations were incubated for 30 min at 42°C, pelleted, and resuspended in 100 µL of YPAD. Resuspended yeast cells were recovered for 2 h at 30°C, washed with 100 µL of $ddH_2O$, resuspended in 10 µL of $ddH_2O$, plated onto selective media (SC–Leu or SC–Trp), and incubated for 3 days at 30°C.

The yN2H assay was performed as described (66) with minor modifications. Briefly, Y8930 with pHM526 (N1/fragment 1 and *LEU2* cassette) or Y8800 + pHM527 (N2/fragment 2 vectors and *TRP1* cassette) were grown overnight in 160 µL SC–Leu or SC–Trp medium at 30°C in a 96-well plate. A positive and random reference set (hsPRS-v2 and hsRRS-v2) in original N2H vectors (66) were grown in parallel. Two protein pairs from the hsPRS-v2 (SKP1-SKP2 and NCBP1-NCBP2) were included in duplicate on every test plate and used as positive controls. Mating was performed by mixing 5 µL of each Y8930 and Y8800 strain in a 96-well plate containing 160 µL YPD medium per well and incubated overnight at 30°C. Strains expressing *L. pneumophila* ORF fusions were also mated with a control strain expressing only fragment 1 (N1) or fragment 2 (N2) to measure the background signal (e.g., N1-X was mated with N2-Y, where X and Y are the proteins tested for interaction, as well as fragment 2 alone). To select for diploid strains, 10 µL of the mating mixture was grown overnight in 160 µL SC–Leu/Trp at 30°C in 96-well plates. Fifty microliters of the diploid selection cultures was transferred into 1.2 mL of fresh medium (SC–Leu/Trp) in deep 96-well plates. Plates were grown overnight at 30°C with shaking; cultures were pelleted (1,800 × *g* for 15 min) and resuspended in 100 µL NanoLuc Assay solution (Promega). The homogenized cell suspensions were transferred into white flat-bottom 96-well plates and incubated for 1 h at RT while protected from light. The luminescence signal was measured using a TriStar2 LB 942 luminometer (Berthold) with a 1 s orbital shake before each measurement and an integration time of 2 s per sample.

For each protein pair X–Y, we calculated an NLR corresponding to the raw luminescence value of the tested pair (X–Y) divided by the maximum luminescence value from one of the two controls (X–fragment 2 or Y–fragment 1) (66). The log-transformed NLR was plotted for human positive and random reference sets previously used with N2H (hsPRS-v2 and hsRRS-v2) and verified iBFG-Y2H pairs (lpPPIs). Fraction detected and confidence clouds (68.3% Bayesian confidence interval) were calculated at each NLR score threshold. Instead of establishing a detection threshold solely reliant on the

hsRRS-v2 pair with the maximum NLR (66), we opted to derive the threshold from the entire distribution of hsRRS-v2 scores. Specifically, we selected a $Z$-score threshold of 2.23, aligning with the next non-null detection value for a data set size of 78, corresponding to a 1/78 hsRRS-v2 detection rate.

## ACKNOWLEDGMENTS

H.O.M. was supported by a CGS-D fellowship from the Natural Sciences and Engineering Research Council of Canada. This work was supported by a project grant (AWE) from the Canadian Institutes of Health Research (PJT-162256). This work was supported by a Wallonie-Brussels International-World Excellence Fellowship (F.L. and G.C.), a Fonds de la Recherche Scientifique (FRS-FNRS)-Télévie Grant (FC31747, Crédit n° 7459421F) (F.L. and J.-C.T.), the Fondation Léon Fredericq (F.L. and J.-C.T.), a University of Liège mobility grant (F.L.), an FRS-FNRS Mobility and Congress funding (no. 40020393) (F.L.), a Josée and Jean Schmets Prize (F.L.), a Herman-van Beneden Prize (F.L.), and an FRS-FNRS-Fund for Research Training in Industry and Agriculture grant (FC31543, Crédit n° 1E00419F) (G.C.). M.V. is a Chercheur Qualifié Honoraire, and J.-C.T. is a Maître de Recherche from the FRS-FNRS (Wallonia-Brussels Federation, Belgium). D.S., A.G.C., N.K., R.L., J.J.K., D-K.K., and F.P.R. were supported by a Canadian Institutes of Health Research Foundation grant (FDN159926).

We thank members of the Roth Lab: Marinella Gebbia for assistance with library construction and Jochen Weile, Natascha van Lieshout, Anjali Gopal, and Nozomu Yachie for bioinformatic advice. Finally, we thank members of the Ensminger Lab: Veronique Cartier-Archambault and Guangqi Zhou for help with cloning; Beth Nicholson, Jordan Lin, and John McPherson for their suggestions and careful reading of the manuscript.

M.L.U. and A.W.E. conceived and designed the screen for effector–effector interactions. H.O.M. constructed the library and performed the inducible BFG-Y2H screen with assistance and training from A.G.C., D.S., and M.L.U. Large-scale gateway cloning was performed by M.L.U., A.G.C., N.K., and R.L., with subsequent robotic cherry-picking by A.G.C., N.K., R.L., D.K., and J.K. Library sequencing was performed by H.O.M., D.S., A.G.C., N.K., and R.L. H.O.M. and M.L.U. analyzed the data with assistance from D.S. M.L.U. performed the Y2H confirmation experiments, M.O.P. assisted M.L.U. with data analysis and network visualization. F.L. performed the yN2H validation experiment with assistance from K.S.F. G.C. analyzed the yN2H data. M.C., J.C.T., M.V., F.P.R., and A.W.E. provided project supervision and advice. M.L.U. and A.W.E. prepared the manuscript with input from other authors.

## AUTHOR AFFILIATIONS

[1]Department of Molecular Genetics, University of Toronto, Toronto, Ontario, Canada

[2]Department of Biochemistry, University of Toronto, Toronto, Ontario, Canada

[3]Donnelly Centre, University of Toronto, Toronto, Ontario, Canada

[4]Lunenfeld-Tanenbaum Research Institute, Sinai Health, Toronto, Ontario, Canada

[5]Center for Cancer Systems Biology (CCSB), Dana-Farber Cancer Institute, Boston, Massachusetts, USA

[6]Department of Genetics, Blavatnik Institute, Harvard Medical School, Boston, Massachusetts, USA

[7]Department of Cancer Biology, Dana-Farber Cancer Institute, Boston, Massachusetts, USA

[8]TERRA Teaching and Research Centre, University of Liège, Gembloux, Belgium

[9]Laboratory of Viral Interactomes, GIGA Institute, University of Liège, Liège, Belgium

[10]Laboratory of Molecular and Cellular Epigenetics, GIGA Institute, University of Liège, Liège, Belgium

[11]Department of Microbiology and Immunology, The University of Melbourne at the Peter Doherty Institute for Infection and Immunity, Melbourne, Victoria, Australia

[12]Department of Computational and Systems Biology, University of Pittsburgh School of Medicine, Pittsburgh, Pennsylvania, USA

## AUTHOR ORCIDs

Harley O'Connor Mount ⓘ http://orcid.org/0000-0003-1401-9178
Malene L. Urbanus ⓘ http://orcid.org/0009-0008-0850-3805
Dayag Sheykhkarimli ⓘ http://orcid.org/0000-0001-8415-6659
Atina G. Coté ⓘ http://orcid.org/0000-0002-0340-9325
Florent Laval ⓘ http://orcid.org/0000-0001-7744-6199
Georges Coppin ⓘ http://orcid.org/0000-0001-7647-3240
Nishka Kishore ⓘ http://orcid.org/0000-0002-9219-4310
Kerstin Spirohn-Fitzgerald ⓘ http://orcid.org/0000-0002-2071-1606
Morgan O. Petersen ⓘ http://orcid.org/0009-0002-2708-0090
Jennifer J. Knapp ⓘ http://orcid.org/0000-0003-3347-4686
Dae-Kyum Kim ⓘ http://orcid.org/0000-0003-4568-8278
Jean-Claude Twizere ⓘ http://orcid.org/0000-0002-8683-705X
Michael A. Calderwood ⓘ http://orcid.org/0000-0001-6475-1418
Marc Vidal ⓘ http://orcid.org/0000-0003-3391-5410
Frederick P. Roth ⓘ http://orcid.org/0000-0002-6628-649X
Alexander W. Ensminger ⓘ http://orcid.org/0000-0003-0824-3704

## FUNDING

| Funder | Grant(s) | Author(s) |
| --- | --- | --- |
| Canadian Government \| Natural Sciences and Engineering Research Council of Canada (NSERC) | | Harley O'Connor Mount |
| Canadian Government \| Canadian Institutes of Health Research (CIHR) | PJT-162256 | Alexander W. Ensminger |
| Wallonie-Bruxelles International (WBI) | | Florent Laval |
| | | Georges Coppin |
| Fonds De La Recherche Scientifique - FNRS (FNRS) | FC31747 | Florent Laval |
| | | Jean-Claude Twizere |
| Fonds De La Recherche Scientifique - FNRS (FNRS) | FC31543 | Georges Coppin |
| Canadian Government \| Canadian Institutes of Health Research (CIHR) | FDN159926 | Frederick P. Roth |

## DATA AVAILABILITY

Strains and plasmids are available upon request. Raw sequencing data are available at the Sequence Read Archive, BioProject ID PRJNA1162088. Raw read counts can be found in Table S7 and Table S8. Barcode assignments can be found in Table S1. A detailed description of how BFG-Y2H sequencing reads are processed is included in the original description of the method [see Note S4 of reference (46)].

## ADDITIONAL FILES

The following material is available online.

### Supplemental Material

**Fig. S1 (mSystems01004-24-s0001.pdf).** iBFG-Y2H barcode representation and correlation of fusion barcode tags.
**Fig. S2 (mSystems01004-24-s0002.pdf).** Retest of iBFG-Y2H interactions with *L. pneumophila* effectors.
**Table S1 (mSystems01004-24-s0003.xlsx).** Open reading frames (ORFs) in the iBFG-Y2H screen.
**Table S2 (mSystems01004-24-s0004.xlsx).** Human positive reference set.

**Table S3 (mSystems01004-24-s0005.xlsx).** *L. pneumophila* ORFs missing from the AD or DB iBFG-Y2H collection.

**Table S4 (mSystems01004-24-s0006.xlsx).** iBFG-Y2H interaction scores for each possible ORF combination.

**Table S5 (mSystems01004-24-s0007.xlsx).** yN2H validation of confirmed iBFG-Y2H interaction pairs.

**Table S6 (mSystems01004-24-s0008.xlsx).** Primers used in this study.

**Table S7 (mSystems01004-24-s0009.xlsx).** Raw count matrices for Up-Up and Dn-Dn barcode combinations from the control (+His) condition.

**Table S8 (mSystems01004-24-s0010.xlsx).** Raw count matrices for Up-Up and Dn-Dn barcode combinations from the Y2H selective condition (-His).

## Open Peer Review

**PEER REVIEW HISTORY (review-history.pdf).** An accounting of the reviewer comments and feedback.

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
