## [Reviewer comments · mSystems]

A comprehensive two-hybrid analysis to explore the *Legionella pneumophila* effector-effector interactome

Harley Mount, Malene Urbanus, Dayag Sheykhkarimli, Atina Cote, Florent Laval, Georges Coppin, Nishka Kishore, Roujia Li, Kerstin Spirohn-Fitzgerald, Morgan Petersen, Jennifer Knapp, Dae-Kyum Kim, Jean-Claude Twizere, Michael Calderwood, Marc Vidal, Frederick Roth, and Alexander Ensminger

Corresponding Author(s): Alexander Ensminger, University of Toronto

Review Timeline:

Submission Date:	July 24, 2024
Editorial Decision:	August 27, 2024
Revision Received:	October 7, 2024
Accepted:	October 13, 2024

Editor: Julia Willett

Reviewer(s): The reviewers have opted to remain anonymous.

Transaction Report:

DOI: <https://doi.org/10.1128/msystems.01004-24>

Re: mSystems01004-24 (A comprehensive two-hybrid analysis to explore the *Legionella pneumophila* effector-effector interactome)

Dear Dr. Alexander W Ensminger:

Thank you for submitting this interesting paper to mSystems. The reviewers highlight the robustness of the approach and the importance of the dataset described in this manuscript. The reviewers have raised some minor points to be addressed in a revision.

Revision Guidelines

Sincerely,
Julia Willett
Editor
mSystems

Reviewer #1 (Comments for the Author):

In this well-written manuscript, O'Connor et al describe a powerful high-throughput genetics-based yeast two-hybrid approach to identify protein-protein interactions (PPIs) between bacterial effectors. *Legionella pneumophila* based on known PPIs between

effector proteins ("metaeffectors") and the Dot/Icm type IV secretion system. Effector-effector interactions are important for Legionella pathogenesis and are notoriously difficult to identify based on their sheer number and lack of virulence phenotypes. Using this new approach, the authors validate several established effector PPIs and identify new PPIs, providing an exciting advance in the field. Most importantly, this technique has the potential to be extended to other pathogens that encode large arsenals of effector proteins, such as *Coxiella burnetii*. I am enthusiastic about this study and have only a few concerns:

1. This approach did not detect several established effector-metaeffector interactions (e.g., LubX-SidH, MesI-SidI, and LegA11-Ceg14). The explanation provided for this discrepancy (lines 325-327) is somewhat underwhelming and additional discussion of false negatives and limitations of Y2H is warranted.
2. Related to the above, the limitations of Y2H necessitate validation of protein-protein interactions by secondary approaches, ideally using purified recombinant proteins *in vitro*. Confidence in this method would be substantially strengthened if at least a few of the novel PPIs were validated using a Y2H-independent approach.
3. Several Dot/Icm T4SS components are integral membrane proteins. Do the authors expect these to have adopted their native conformation in yeast?

Reviewer #2 (Comments for the Author):

In the article « A comprehensive two-hybrid analysis to explore the Legionella pneumophila effector-effector interactome » O'Connor Mount and colleagues developed a previously published Y2H screen for effector-effector interactions further. They systematically and deeply analyze the protein effector interactome and the interaction of the effectors with the type IV secretion apparatus proteins and the interaction of the type IV apparatus proteins among them in the pathogen Legionella pneumophila. This is a particularly challenging task, as *L. pneumophila* is the pathogen with the highest number of secreted effectors known, to date more than 330 have been identified and confirmed.

Many of these effectors are toxic when expressed in human cells or yeast, hindering many analyses. Here the authors circumvented this problem by modifying the BFG-YSH vector by replacing the constitutive promoter with a copper-inducible promoter which allowed to avoid expressing the Legionella proteins during haploid growth. This elegant method allowed to identify new effector-effector interactions. To validate these interactions another elegant assay using the nanoLuc two-hybrid assay was set up, again with copper-inducible promoters allowing to validate the entire dataset. Finally, the authors present an effector-effector interaction network that shows that effector-effector interplay is common.

The work presented here seems to be well done and is clearly described. The resulting dataset will be a valuable recourse for all researchers in the Legionella field that can now test the newly identified effector-effector interactions and the deduced functional hypothesis; Furthermore, the method described here is well suited to be used with other pathogens that secrete a number of effectors such as *Coxiella*, *Chlamydia* or others.

I have no major concerns but would suggest change/adding some of the references in the introduction to be more accurate.

Page 6, line 98 reference #11 is from 2002 and there are other very comprehensive, newer reviews. I suggest adding "Newton et al, Clin Microbiol rev, 2010 and Mondino et al., Annu Rev Pathol 2020

Page 6, Line 104 the reference was not correctly formatted. In addition, I suggest citing the original paper that describes *L. pneumophila* the first time here: McDade, 1977, N Engl J Med

It is a little difficult to follow in the text how many new effector pairs have been identified how many are already known etc as most of the time words like "several" are used (e.g. Line 277, line 281 etc...) Please replace these with exact numbers

Line 360-361 please delete the double words "to restrict"

We thank the reviewers and the editor for their kind words and thoughtful comments on the manuscript. We are excited about the prospects of providing this resource to the community through mSystems and have addressed each of their specific comments and suggestions below.

Reviewer #1 (Comments for the Author):

In this well-written manuscript, O'Connor et al describe a powerful high-throughput genetics-based yeast two-hybrid approach to identify protein-protein interactions (PPIs) between bacterial effectors. Legionella pneumophila based on known PPIs between effector proteins ("metaeffectors") and the Dot/Icm type IV secretion system. Effector-effector interactions are important for Legionella pathogenesis and are notoriously difficult to identify based on their sheer number and lack of virulence phenotypes. Using this new approach, the authors validate several established effector PPIs and identify new PPIs, providing an exciting advance in the field. Most importantly, this technique has the potential to be extended to other pathogens that encode large arsenals of effector proteins, such as Coxiella burnetii. I am enthusiastic about this study and have only a few concerns:

Many thanks to the reviewer for the nice comments and enthusiasm about the study.

1. This approach did not detect Several established effector-metaeffector interactions (e.g., LubX-SidH, MesI-SidI, and LegA11-Ceg14). The explanation provided for this discrepancy (lines 325-327) is somewhat underwhelming and additional discussion of false negatives and limitations of Y2H is warranted.

Thank you for this thoughtful comment. We have added the following text to the Discussion to place these “false negatives” within the context of the established limitations of protein-protein interaction detection methodologies. Specifically, we now write (Lines 375-387):

Beyond the positive hits of the iBFG-Y2H screen, a number of known interactions were not captured by the iBFG-Y2H screen. These include the metaeffector LubX and its effector target, SidH, as well as metaeffector-effector pairs that we previously identified in targeted Y2H interrogation: SidP-MavQ, LupA-LegC3, and SdbC-SdbB (18). There are likely several reasons for this, including: known sensitivity limitations (e.g. not optimized for membrane protein interactions, interactions depending on specific subcellular localization or post-translational modifications) (112), the confounding effects of yeast toxicity despite our inducible plasmids (18, 39), and protein instability (e.g. SidH is polyubiquitinated by LubX and subject to proteosomal degradation) (16). This rate of “false negatives” is not unexpected as it is well-established that any one binary PPI assay is likely to detect only $\sim 1/3^{\text{rd}}$ of high-confidence, benchmarked interactions (66, 67, 70, 113–118). Altogether, this cautions against interpreting lack of detection as either a lack of interaction or a unique limitation of the assay as performed. Instead, it suggests that more effector-effector interactions are likely to be revealed through the application of several complementary, orthologous methodologies (66).

2. Related to the above, the limitations of Y2H necessitate validation of protein-protein

interactions by secondary approaches, ideally using purified recombinant proteins in vitro. Confidence in this method would be substantially strengthened if at least a few of the novel PPIs were validated using a Y2H-independent approach.

As mentioned in the reworked Discussion paragraph highlighted above, a number of groups (see most recently S. G. Choi *et al.*, *Nat. Commun.* 10, 3907 (2019)) have shown that several orthogonal assays are required to fully characterize a complex interaction network. Each binary interaction method (e.g. Y2H, N2H, AP-MS) can only capture a subset of the protein-protein interactions. Validation of large screens such as this are generally done by using an orthologous method to compare the performance of the assay against known positive reference sets (PRS) (see Fig 2 and 3). Using yN2H (an orthologous and independent method), we confirmed 16 Y2H interactions of which 9 are novel (see Table S5). Performance of the screen with respect to effectors was comparable to the PRS (Fig 3), which, given the limitations of any one PPI methodology, is generally interpreted as validating the overall performance of the primary screen.

We have modified the description in the Discussion to clarify these results in lines 334-338:

A subset iBFG-Y2H PPIs were detected in the orthologous NanoLuc Two-Hybrid (N2H) assay (66) at a rate of 48.5% (Fig. 3). This is better than the detection rate for a high-quality positive reference set, indicating that the verified iBFG-Y2H set is a high-confidence dataset – with 16 PPIs captured in both assays of which 9 are novel (see Table S5).

Further supporting the protein interactions captured using the iBFG-Y2H screen, we previously performed affinity purification coupled with mass spectrometry using bead-immobilized PieF (H. O'Connor-Mount, *Biorxiv*, 2022). In this preprint, AP-MS showed that CNOT7, CDK4 and LSM3 copurified with PieF; interactions that were also identified in using the iBFG-Y2H assay and showed that the PieF-CNOT7 interaction can be captured using the bacterial two-hybrid method, BACTH. Similarly, we have previously reported on direct interactions between RavJ/LegLI using gel filtration, Y2H, LUMIER, and co-crystallization (M. L. Urbanus *et al.*, *Mol. Syst. Biol.* 12, 893 (2016)) and MavE/YlfA using Y2H and LUMIER (M. L. Urbanus *et al.*, *Mol. Syst. Biol.* 12, 893 (2016)). For other pairs, we view this report as a Resource Report (submitted as such), with the expectation that other investigators will deploy a variety of orthologous methodologies to study the interactions and functional outcomes of the novel complexes revealed by our effector-wide screen.

3. Several Dot/Icm T4SS components are integral membrane proteins. Do the authors expect these to have adopted their native conformation in yeast?

We have not confirmed if these proteins have adopted their native confirmation in yeast, but we indeed observe Y2H interactions with integral membrane proteins, indicating they are not merely degraded and possibly are integrated into a membrane.

To place this into context, while the Y2H assay is not optimized to capture membrane protein interactions, the Human Reference Interactome (K. Luck *et al.*, *Nature.* 580, 402–408 (2020),

<http://www.interactome-atlas.org/>) which is the repository for several comprehensive human Y2H screens, lists several membrane protein interactions. Some of these interactions have also been demonstrated using other methods, such as SNARE proteins VAMP3 and STX4 (F. Paumet *et al.*, *J. Immunol.* **164**, 5850–5857 (2000); J. Polgár, S.-H. Chung, G. L. Reed, *Blood.* **100**, 1081–1083, (2002)).

Reviewer #2 (Comments for the Author):

In the article « A comprehensive two-hybrid analysis to explore the Legionella pneumophila effector-effector interactome « O'Connor Mount and colleagues developed a previously published Y2H screen for effector-effector interactions further. They systematically and deeply analyze the protein effector interactome and the interaction of the effectors with the type IV secretion apparatus proteins and the interaction of the type IV apparatus proteins among them in the pathogen Legionella pneumophila. This is a particular challenging task, as L. pneumophila is the pathogen with the highest number of secreted effectors known, to date more than 330 have been identified and confirmed.

Many of these effectors are toxic when expressed in human cells or yeast, hindering many analyses. Here the authors circumvented this problem by modifying the BFG-YSH vector by replacing the constitutive promoter with a copper-inducible promoter which allowed to avoid expressing the Legionella proteins during haploid growth. This elegant method allowed to identify new effector-effector interactions. To validate these interactions another elegant assay using the nanoLuc two-hybrid assay was set up, again with copper-inducible promoters allowing to validate the entire dataset. Finally, the authors present an effector-effector interaction network that shows that effector-effector interplay is common.

The work presented here seems to be well done and is clearly described. The resulting dataset will be a valuable recourse for all researchers in the Legionella field that can now test the newly identified effector-effector interactions and the deduced functional hypothesis; Furthermore, the method described here is well suited to be used with other pathogens that secrete a number of effectors such as Coxiella, Chlamydia or others.

I have no major concerns but would suggest change/adding some of the references in the introduction to be more accurate.

We thank the reviewer for these kind comments and thoughtful suggestions. We have made the suggested changes as outlined below.

Page 6, line 98 reference #11 is from 2002 and there are other very comprehensive, newer reviews. I suggest adding "Newton et al, Clin Microbiol rev, 2010 and Mondino et al., Annu Rev Pathol 2020

Done.

Page 6, Line 104 the reference was not correctly formatted. In addition, I suggest citing the

original paper that describes L. pneumophila the first time here: McDade, 1977, N Engl J Med

Done.

It is a little difficult to follow in the text how many new effector pairs have been identified how many are already known etc as most of the tile words like "several" are use (e.g. Line 277, line 281 etc...) Pleas replace these with exact numbers.

Thank you for the suggestion to add more specificity, we agree that it helps make the manuscript's findings much easier to follow. We have replaced words like "several" with specific numbers for each class of hits throughout.

Line 360-361 please delete the double words "to restrict"

Thank you for the catch. Done.

Re: mSystems01004-24R1 (A comprehensive two-hybrid analysis to explore the *Legionella pneumophila* effector-effector interactome)

Dear Dr. Alexander W Ensminger:

Your manuscript has been accepted, and I am forwarding it to the ASM production staff for publication. Your paper will first be checked to make sure all elements meet the technical requirements. ASM staff will contact you if anything needs to be revised before copyediting and production can begin. Otherwise, you will be notified when your proofs are ready to be viewed.

Sincerely,
Julia Willett
Editor
mSystems